# “No Child or Adult Would Ever Probably Choose to Have 16 Teaspoons of Sugar”: A Preliminary Study of Parents’ Responses to Sugary Drink Warning Label Options

**DOI:** 10.3390/nu14194173

**Published:** 2022-10-07

**Authors:** Caroline Miller, Joanne Dono, Kathleen Wright, Simone Pettigrew, Melanie Wakefield, John Coveney, Gary Wittert, David Roder, Sarah Durkin, Jane Martin, Kerry Ettridge

**Affiliations:** 1School of Public Health, The University of Adelaide, Adelaide 5000, Australia; 2Health Policy Centre, South Australian Health and Medical Research Institute, Adelaide 5000, Australia; 3School of Psychology, The University of Adelaide, Adelaide 5000, Australia; 4Food Policy, The George Institute for Global Health, University of New South Wales, Sydney 2042, Australia; 5Centre for Behavioural Research in Cancer, Cancer Council Victoria, Melbourne 3004, Australia; 6School of Psychological Sciences, The University of Melbourne, Melbourne 3010, Australia; 7College of Nursing and Health Sciences, Flinders University, Adelaide 5042, Australia; 8Freemasons Centre for Male Health and Wellbeing, South Australian Health and Medical Research Institute, and The University of Adelaide, Adelaide 5000, Australia; 9Centre for Nutrition and GI Diseases, South Australian Health and Medical Research Institute, Adelaide 5000, Australia; 10Cancer Epidemiology and Population Health, University of South Australia, Adelaide 5000, Australia; 11Obesity Policy Coalition and Alcohol and Obesity Policy, Cancer Council Victoria, Melbourne 3004, Australia

**Keywords:** warning labels, sugary drinks, sugar-sweetened beverages, parents, children, adolescents, qualitative

## Abstract

Front-of-pack (FoP) warning labels are a viable policy option with the potential to inform consumer choice and assist in reducing sugar-sweetened beverage (SSB) consumption as part of a multi-faceted approach. This study explored parents’ perceptions and understanding of a range of SSB warning labels. Focus groups *(n* = 12) with 82 parents of school-aged children were conducted, stratified according to education level, sex and location. Health effects, exercise equivalents, sugar content (teaspoons in text and pictograms, “high in”) and energy content labels were shown. Through thematic analysis we identified three themes. Theme 1 related to perceptions of effectiveness of labels, underpinned by four subthemes: perceptions of labels as credible, informative and useful, personally relevant and having the potential to change be haviour. Theme 2 related to participants finding opportunities for self-exemption (e.g., physically active) and message rejection (e.g., misinterpretation). Theme 3 encompassed the potential negative consequences of some labels (e.g., body image concerns). The text teaspoons label was perceived most favourably across all themes, with minimal negative issues raised. These results provide in-depth insight into potential responses to labelling as a policy intervention, providing important guidance for the development of labels to ensure optimal message content and framing for future testing and subsequent implementation.

## 1. Introduction

Sugar-sweetened beverages (SSBs) are non-alcoholic beverages with added sugar, such as soda (soft drink), fruit drinks, sports drinks, vitamin waters and energy drinks [1]. SSBs are a major source of added sugar, are often consumed in high volumes, are limited in satiety and nutritional value, and displace other nutrient-rich foods in the diet [2]. High consumption of SSBs is problematic due to recognised contributions to childhood and adult weight gain and obesity, as well as a number of other adverse health effects and risks, specifically, dental caries, type 2 diabetes and cardiovascular disease risk [3,4,5,6,7,8]. Globally, the burden of disease attributed to SSBs has been estimated at 184,000 deaths, with the lowest rate occurring in low-income countries (5%), 71% in middle-income countries and 24% in high-income countries [9]. Within Australia, SSBs are the largest source of free sugars [10]. Over half of Australians (52%) exceed World Health Organisation recommendations that free sugar intake should be limited to less than 10% of energy intake, with 90% exceeding the conditional recommendation to limit free sugar intake to less than 5% [2,10]. National Australian data indicate SSBs are consumed by children as young as two years and consumption peaks in adolescence, with adolescent boys aged 14–17 years among the highest consumers [11].

Regular consumption and overconsumption of SSBs during childhood can set up dietary habits that can continue into adulthood, with consequential health risks [12,13,14,15]. Primary caregivers are important gatekeepers to young children’s consumption of discretionary food and drinks, with most sugary drinks consumed within households [16]. While adolescents are in a period where behaviour is more autonomous and they may purchase their own food and drink, SSB consumption still most often occurs in household environments [16,17]. It is therefore important to have policies in place to support parents’ healthy decisions and purchasing behaviours to reduce the availability of SSBs to children and adolescents within the home and promote healthy environments for children and adults.

One policy option with potential to assist consumers in making informed choices at the point of purchase and consumption, as part as a comprehensive multi-faceted approach, is warning labels on front-of-pack (FoP) [18,19]. The evidence that warning labels can improve health outcomes, e.g., reduce SSB consumption and purchasing, is increasing. Meta-analyses, systematic reviews and a scoping review of quantitative studies provide good evidence of these outcomes [20,21,22,23,24]. This research has also suggested that consumer reactions vary with different label types. For example, the meta-analysis indicated that health effects warning labels may be more effective than nutrient warnings (“high in”) in reducing consumption intentions. A scoping review found nutrient warnings and health effects were both perceived highly on different attributes. For example, nutrient (“high in”) warning labels were perceived as easy to understand, helpful and discouraged unhealthy purchasing, however, the health effects labels provided more information about relative healthiness of products to consumers.

Since Chile first implemented FoP labels in 2016 to warn consumers of products that were “high in…” nutrients such as saturated fats, salt and/or sugar, over 30 countries have now implemented similar policies and labelling systems [25,26,27]. While Australian adults show high community support for warning labels and adolescents report low opposition [28,29,30], little has been done to progress this potential policy option.

In Australia, a voluntary FoP health star rating system is in place where companies can elect to display a health star rating on foods and beverages to indicate the product’s healthiness. Uptake of this system has been low overall, particularly for beverages, with greater uptake for healthier beverages [31,32]. A recent ‘real stakes’ study indicated that health stars have the potential to reinforce the effects of warning labels on beverages [33]. Given the voluntary nature of the current health star system, the addition of mandatory warning labels to enable consumers to quickly identify unhealthy beverages is worthy of further consideration.

As previously mentioned, quantitative studies have provided a good indication that warning labels as a policy option are worthy of consideration, however they do not offer detailed insight into the underlying perceptions and understanding of labels among different groups of consumers. This insight can be ascertained from studies that utilise qualitative approaches. However, most qualitative studies [18,34,35,36,37,38] that have explored perceptions of FoP labels that have included warning labels have been conducted in South American countries, explored reactions to health messages or labels more broadly (e.g., labels or messages applicable to all food and beverages rather than SSB-specific labels or health messages), and/or have been conducted with small, focused samples. While heterogeneity across these qualitative studies limits the ability to draw comparisons, perceptions of credibility and the need for simple and easy-to-interpret messages were recurring results among different participant groups. Furthermore, the use of pictures or icons appeared to enhance participants’ understanding compared to text-only labels.

There have been two larger qualitative studies conducted with young adults in different settings (US and Australia). These studies have explored reactions to a broader range of potential warning labels on SSBs. The mixed-methods US study of college students’ (n = 86) perceptions of a range of potential SSB warnings (on-product and point of sale) found for text-only messages, attribution to the college health centre increased perceived effectiveness by increasing credibility [39]. Brevity and clarity of the messages were valued, as were warnings with a picture/icon. Preferences were generally for label designs that drew attention to the warning element (e.g., via marker word, shape, or exclamation symbol), and for those that conveyed sugar content in an easy-to-understand manner. The qualitative study of young Australian adults (n = 105) explored reactions to a range of SSB warning labels depicting sugar content, energy content (calories, KJs, % energy intake), health effects, and exercise equivalent information. This study found young adults’ perceptions of the potential effectiveness of labels were highly related to perceptions that a label was: useful, credible, personally relevant, robust to self-exemption, elicited negative emotive reactions, and likely to prompt behaviour change [40]. Labels communicating sugar content in an easy to interpret way (via teaspoons in text or pictogram) were perceived to be the most effective overall, and were regarded as having the most potential to curb consumption and purchasing behaviour. These studies offer new insights into consumer reactions to a range of labels, however they were specifically focused on young adults or college students, limiting the generalisability of results to other important consumer groups.

Our current insight into consumer reactions to and understanding of different warning label types specific to SSBs is limited. There have been very few studies that have purposively included parental perspectives [18,34,35,36,37,38]. This is a notable gap in existing research, as parents are an important group of consumers as gatekeepers to children’s consumption. With many countries contemplating labelling policies, it is important to ascertain perceptions and understanding of a range of different warning labels among important groups of consumers in different settings and contexts. This can be achieved by employing a qualitative approach to enable insight into parental understanding and perceptions of warning labels that cannot be ascertained with quantitative methodology. The objective of this study was to conduct a preliminary exploration to determine parents’ perceptions and understanding of a range of warning labels (21 in total) from their own perspective and their perceptions of their children’s understanding and potential reactions.

## 2. Materials and Methods

### 2.1. Ethical Approval

The Human Research Ethics Committee of The University of Adelaide approved this research.

### 2.2. Design

This was a focus group study with a qualitative descriptive design to enable detailed insights into parental reactions to warning labels [41,42]. A graphical depiction of the research process and design is available from Appendix A (see Appendix A). An external company, MMresearch (Melbourne, Victoria), was commissioned to recruit participants and moderate the focus groups. Twelve focus groups of 3–8 participants were conducted with parents of school-aged children. To obtain a range of parental views, the groups were stratified by school grade of child (primary school grade 1–6/secondary school grade 7–10); location (metropolitan/regional); sex (male/female); and parental education (no tertiary/tertiary). The metropolitan groups were held in a large Australian city (Melbourne, Victoria) and regional groups were held in a regional town (Ballarat, Victoria).

### 2.3. Participants

MMResearch conducted participant recruitment through professional recruitment companies via their existing participant pools. Potential participants were first contacted by email. Those interested completed a brief telephone interview to check eligibility: being a parent (aged 18 or older) of a child in school grades 1–10 (residing with the parent at least some of the time); being the main grocery buyer for the household; purchasing, personally consuming, and child(ren) consuming, sugary drinks ‘at least weekly’; and they or close friends/family do not work for the beverage industry. Informed consent was gained both in writing and verbally at the beginning of each focus group. Participants were reimbursed 80 AUD for their time.

### 2.4. Focus Groups

The moderator conducted an initial discussion regarding participants’ beverage consumption to develop rapport, ensure people were at ease and that they understood the researchers were interested in their experiences and perceptions. Participants were asked to discuss their own consumption and what they provided for their children. A series of warning labels was then shown to participants on an electronic screen. The moderator probed to elicit responses to and comparisons between labels. Probes included initial reactions to and interpretations of each label and whether the label: taught something new, was believable, made participants stop and think, was considered relevant, impacted purchase or consumption intentions, and caused participants to feel uncomfortable about their own or their children’s sugary drink consumption. These prompts were based on established measures of perceived effectiveness from the tobacco and SSB literature [21,43]. The core research team (JD, CM, KE) viewed groups via a live link where possible, and regularly debriefed with the moderator to ensure preconceptions were not imposed on participants and to discuss arising concepts and interpretations.

### 2.5. Warning Labels

The warning labels were presented to participants in sets (see Table 1 for labels and sets) and displayed on a presentation screen in plain black and white (no colour). The labels developed and selected for inclusion in this study were based on those used in the SSB warning label literature and labels implemented in real life [25,40,44,45,46,47]. Some modifications were made to some labels to increase participant understanding based on results of a qualitative study of young adults [40]. For example, the word “extra” was included in exercise warning labels to clarify it was in addition to everyday activity. Labels depicting calories were also referred to as “avoidable” to enhance participant understanding that these calories were surplus to need.

### 2.6. Data Analyses

Due to limited facilities in the regional location, regional groups (6 groups) were audio recorded and metro groups were video recorded (6 groups). Recordings were used for transcription. The core research team (CM, JD, KE, KW) reviewed all groups either live or via recordings. The approach to analysis was thematic analysis according to Braun and Clarke [48]. This involved one researcher (KW) reviewing all transcripts, and conducting coding to nodes within label sets using NVivo software [49]. Both inductive and deductive coding was undertaken, with the initial coding structure guided by a framework derived from the discussion guide, the previous coding framework used for a study of young adults, as well as the core research teams’ initial discussion of the groups. Themes were identified and developed across the data set in consultation with core research team (CM, JD, KE). Where differences arose, reflections of transcripts and group observations were undertaken until consensus was obtained. Quotes are presented within the text to exemplify participants’ discussion relating to the themes and subthemes. Quotes are presented with the group descriptors of group location (Metropolitan (Metro) or regional (Reg)), group sex (Male or Female), group education level (low (Low) or medium–high (H)), and child school grade level (primary and secondary).

## 3. Results

The total number of participants was *n* = 82 across *n* = 12 groups. Table 2 provides numbers per group by location, parent education level and sex and child school level.

Three main themes were identified, with constituent subthemes described within Theme 1 (see Table 3).

### 3.1. Perceptions of Label Effectiveness

#### 3.1.1. Credibility

Across all assessed labels, perceptions of credibility and seriousness were facilitated by the marker word ‘Warning’ and label shape. The rectangle shape was perceived as reminiscent of tobacco warnings. The hexagon shape facilitated perceptions of seriousness as it was a similar shape to a stop sign or danger symbol.

“I looked at that and the first thing that came to mind was that’s what’s on a packet of cigarettes!”(Metro, F, H, S)

“Looks like a sign that should be on the back of a truck, Warning! You know explosives enclosed or something”(Reg, F, L, S)

Labels perceived as factual and correct were considered credible, e.g., the teaspoons of sugar labels were perceived as factual and hard to argue with or discount.

“It’s got 16, they’ve done the tests”(Metro, M, L, P)

In contrast, participants objected to the word “causes” in the health warning labels, commenting not all consumers acquired these diseases, and there were other causes of these health outcomes. Diabetes was challenged less than obesity, consistent with the perception of many participants that diabetes has a direct link to sugar consumption and is more serious than obesity. Tooth decay was generally accepted, though some maintained not all consumers have tooth decay or bad teeth. Overall, there was a strong preference for the wording “contributes to” rather than “causes”.

“Doesn’t cause obesity! Just because you drink sugar doesn’t mean you’re guaranteed you’re going to be obese. I consume a lot of sugary drinks and I’m half-anorexic!”(Metro, F, L, P)

“It’s not the cause, the word causes in there to me is not right. It contributes, but not causes. I think it depends on the quantity too, like my kids have a little bit of soft drink a week and I don’t think that’s going to give them obesity or diabetes or anything”(Reg, M, H, P)

#### 3.1.2. Informative and Useful

All participants had a good understanding of teaspoons as a metric, and this information was perceived as valuable and as having potential to inform choice. It captured interest regardless of whether it was considered new (as it was for many participants) or familiar, e.g., from dentists or schools (as noted by some participants). Many participants commented that children would quickly and easily conclude that consuming so much sugar is not healthy, this information made it easy to add up sugar intake across a day or week and compare across drinks, and it was easy to visualise the quantum of sugar.

“It’s also about being informed, too. That’s a blatant way of saying what a nutritional information label might say. With that, everyone understands what a teaspoon is and you go 16 isn’t good”(Reg, M, L, S)

“… if I said to my daughter, okay, have that cup of tea and I’m going to put 16 teaspoons of sugar in that and then drink it, no way, oh, it would put her off. So it’s something that you can translate”(Metro, F, H, S)

Most participants considered health effects information as well-known with limited potential to capture attention or provoke thought.

“But I think I already know those things … So I’m not going to pay much attention to it”(Metro, F, H, S)

“I look at that [obesity message] and it has no impact on me … because all of us know that it causes obesity, but we still keep doing it, so that sign to me, I looked at that and had no feeling about it. No impact”(Reg, F, L, S)

Many participants perceived the exercise equivalent label information as interesting, simple and relatable for themselves and their children. However, many did not interpret the activity as *additional* to current activity, with more potential to capture attention for the small proportion who perceived it as additional exercise.

“My job is already walking and it’s more like an office job, so I’d be like yeah that’s okay, I do a lot of walking anyway”(Metro, M, L, P)

“I reckon my kids would say well I walk enough in the day as it is so that’s it”(Metro, M, L, P and S)

Participants perceived the “high in sugar” label as simple, however, they also indicated this information was well-known and was too generic or abstract to be meaningful or useful.

“I think we already know it’s high in sugar”(Metro, F, L, P)

“And what is high in sugar, like you need some context around it”(Reg, M, H, P)

Most participants did not understand the calorie information or the % RDI (recommended daily intake) labels and indicated their children would not either. Only the small proportion of participants who could interpret the contents labels (cals/KJs) perceived them as useful.

“Too scientific, would mean absolutely nothing to me and nothing to the kids”(Metro, M, L, P and S)

“How do you know how many calories we need in a day?”(Metro, F, L, P)

#### 3.1.3. Personal Relevance

Perceptions of personal relevance for themselves or their children were related to participants’ personal health priorities. Participants generally perceived health effects labels to have low relevance, except for a small proportion with personal connections to the health effect and/or those with very active, sport-focused children.

“I think that they would actually look at it [health warnings] … they play sport, they’re healthy”(Metro, F, H, P and S)

“I think obesity and diabetes. I think my son’s more switched on with diabetes because my husband has Type II and he sees the effects of that. My daughter because of the body image, being 14”(Metro, F, H, P and S)

Exercise messages varied in personal relevance depending on individuals’ physical activity preferences.

“I think when you put the kilometres [more impactful], whereas minutes you might be doing minutes at a slow pace or fast pace”(Metro, M, L, P)

“I think that [running] would have more of an impact for most people because then they’d think, ‘Oh now I’ve got to run!’. Walking is much easier”(Reg, M, H, P)

It was noted by some that this information may be ignored (by adults and children) if the exercise depicted was not considered personally relevant.

“I just think stuff it I can’t run … I wouldn’t put the drink down”(Reg, F, L, P and S)

“Anyone can walk…whereas running I’d just go I can’t do that, I’ll just drink the drink”(Reg, F, L, P and S)

Teaspoons of sugar labels (text or pictogram) were perceived as more universally relevant, such that the amount of sugar was considered excessive for children, adolescents and parents alike. Varying activity levels or being health conscious were not raised in the context of these labels.

“I think that the 16 teaspoons is like anyone can relate to the 16 teaspoons”(Metro, M, L, P and S)

#### 3.1.4. Influence on Consumption Behaviour

Perceptions of whether labels would be likely to change behaviour (own, children’s or others’) were strongly related to perceptions described in the previous subthemes. The teaspoons of sugar labels were perceived strongly on all previously discussed attributes, and were consistently described as being likely to change behaviour. These labels prompted participants to consider the implications for themselves and their children, with many indicating they would not consume a beverage containing 16 teaspoons of sugar. Others felt it would prompt them to switch to a lower sugar drink. This appeared to be strongly driven by the shock value of the teaspoons message, initially evident for the text version that was shown first in the groups.

“That’s lifechanging, I’m not being overly dramatic, absolutely, that is life changing”(Metro, M, L, P)

“I’m just thinking that now in my head, how many teaspoons of sugar am I having a day? … That’s a lot. That’s shocking for me. Because I don’t think of it like that.”(Metro, F, L, P). (consumes five cans of coke a day)

“I wouldn’t touch it and I wouldn’t put it in my mouth. That amount of sugar”(Metro, F, H, P and S)

The teaspoon pictogram labels generally received similar responses, for parents themselves and in relation to their children; albeit reactions were not as strong overall.

“No child or adult would ever probably choose to have 16 teaspoons of sugar, into their body, they would just not want to do that”(Metro, M, L, P)

A small number of participants indicated that teaspoons information would not change their choice of beverage, but it would make them think about their options, and may influence behaviour over time. Parents tended to express greater concern for their child’s consumption than their personal consumption and indicated an intention to reduce their child’s intake based on the teaspoons information. For those with young children, this was through parental gatekeeping, e.g., changing what is brought into the home.

“Yeah well, you’d think twice whether I should be giving it to them or limiting how much I’m going to give”(Metro, M, L, P)

Many of the parents of older children and adolescents also thought that the teaspoons information would influence their child to choose a lower-sugar option.

“And I think it’s good education for the kids as well, because I don’t think they fully understand how much sugar is in soft drinks … But if they can read that, my daughter would definitely be like, “oh god, I’m not having that”, you know”(Metro, F, H, S)

However, a minority indicated that their adolescent would be unlikely to change their behaviour on seeing the teaspoons of sugar information.

“My kids do care, but… You know they’ll still have the sugary drinks”(Reg, F, H, P)

Other labels elicited mixed perceptions regarding the potential impact they may have on beverage consumption. Some parents commented these labels may stimulate thought regarding options they allow for their children, but many felt that adolescents would ignore the information. For example, the health effects and exercise labels were generally considered to have low immediate impact and were considered unlikely to change behaviour, although some indicated there might be potential to reconsider their choice over time.

“They’re still going to drink it, but as they get older and wiser I think it’s that repeat looking at things. It’s the same for me if I keep looking at it”(Metro, F, H, P and S)

For the minority of participants who fully understood the calorie and % RDI sugar labels, there was potential to motivate change to a lower sugar beverage or water. However, the 12% RDI label was perceived as having potential to increase consumption. Twelve was perceived as a low number, and energy was perceived as a good thing that is required by the body daily to survive.

“That to me says, “This is great, it’s giving you 12% of the energy you need, have more!””(Metro, F, L, P)

### 3.2. Opportunities for Self-Exemption and Message Rejection

Participants found opportunities for self-exemption to some labels through questioning credibility, resisting the message “tone”, misinterpreting, and nominating remedial actions. As previously noted, participants questioned the credibility of the health effects messages. Some participants also took exception to message tone of the health effects messages, perceiving these labels as authoritarian. This was particularly noted when the health effect of obesity was presented with the educative message “16 teaspoons of sugar” (see column 2, Table 1). Most commonly, participants noted that the health statement could diminish or detract from the 16 teaspoons statement or enable self-exemption, although some perceived the 16 teaspoons to strengthen the health statement.

“No, I just think it’s the government telling me what to do. So if I want to drink a Coke, I’ll drink a Coke. The same as if I want to smoke, I would smoke”(Reg, M, H, P)

“Going back to the second thing there, the excess sugar causes obesity. We all know that, and I feel like that’s been thrown in our face, and when you get something thrown in your face, you are more likely to go, yeah whatever!”(Reg, F, L, S)

A frequent argument across all labels except the teaspoons labels was that everyone’s bodies are different, enabling participants to minimise or discount some of the label information. For example, the content/energy labels appeared to provide opportunity for self-exemption for those who may not consider themselves “average”.

“What have they based that on… you know what age of the person, weight of the person, fitness level?… we are all going to have different intakes, based on what we eat and drink throughout the day”(Reg, M, L, P and S)

Similar indications were also observed for health effects messages.

“There’s more factors to it. As you said, you know, how much you drink, or what your activity is like”(Reg, M, L, P and S)

Perceived scope for remedial action also offered opportunity for self-exemption. Some participants commented that they or their child already engage in so much physical activity the exercise warning would not apply to them. They noted that adolescents would self-exempt from exercise equivalents and the health effect of obesity, due to being of thin physique and/or active enough to offset these risks. Participants also noted brushing teeth as a remedial action to offset tooth decay.

“Doesn’t really make me think twice about it. I walk around all day all the time anyway. It doesn’t sort of stand out and say this is really bad for you”(Reg, M, H, P)

“My kids would say, well I walk enough in the day as it is, so that’s it. So I walk enough, so I’ll walk it off anyway”(Metro, M, L, P and S)

Participants’ comments generally indicated the teaspoons information to be resistant to self-exemption, with some exceptions. When shown as a pictogram (picture of teaspoon of sugar X 16; see column 6, Table 1), a minority questioned the substance on the teaspoon or expressed confusion regarding the meaning of X (intended to be a multiplication symbol). While this discussion slightly detracted from the impact of the message, some participants still indicated a preference for the pictogram teaspoons message, particularly for children as it involved less reading. However, others felt the pictogram was not required for adults and did not increase impact of the initial text message.

“I’m just, maybe I’m negative, I don’t know. But I just think that looks like 16 times better than that drink over here. That’s what it looks like…”(Metro, M, L, P and S)

“Although it doesn’t say it’s sugar, it could be salt, you know”(Reg, F, H, P)

#### 3.2.1. Potential Negative Implications

Some concerns were raised in relation to each label set. Age appropriateness was an issue for some parents, particularly for health effects. Tooth decay was regarded as a simple issue to raise with younger children, while diabetes was considered more complex and more appropriate for discussion with older children or adolescents. Body image concerns were raised in response to the obesity label.

“Because it really depends on the age [when to discuss health issues], I think we started off with tooth decay, then we moved to like the obesity thing and the cancer. And then the diabetes more recently”(Reg, F, H, P)

“For a 6 year old I would only talk to them about tooth decay. She wouldn’t understand diabetes, and I don’t want to talk about body image yet [equating obesity label to body image]”(Reg, F, H, P)

“Obesity to me is a really bad word to use … That word to me is a very dangerous word, more than a lot of other words are. That word to me on a warning label is not good for kids”(Reg, F, L, P and S)

Exercise equivalents were acknowledged as having little impact on children who were very active or of lean physique. Some parents with highly active children suggested that their child may use the label information to argue for increased consumption (e.g., as a reward for activity).

“But the kids do a lot of running anyway, so they run around the playground and all that kind of stuff. The kids say, “well I’ve done 20 min today. You got a Coke?””(Metro, M, L, P and S)

There were concerns that the focus on sugar content may lead to misinterpretation among young people. For example, a drink with lower sugar, such as an energy drink or sport drink may be perceived as “healthier” based on sugar content, despite the other harmful ingredients that may be present. Some participants noted that showing drinks with different sugar content would prompt them to choose drinks with a lower overall quantum of sugar, but not necessarily water.

“A kid will look at a Coke, look at a PowerAde, they’ll be, “stuff it I’ll have a PowerAde”. PowerAde’s no good for them though. But it’s just got a little bit less sugar that’s all. But it makes up for it in other ways”(Metro, M, L, P and S)

“It would affect the way I choose my drinks. Like I said, it makes you sit down and think well obviously out of that I’d be drinking the PowerAde, I wouldn’t bother grabbing any others”(Reg, M, H, P)

#### 3.2.2. Further Considerations

Within label discussions participant’s responses indicated that clarifications added to messages (based on recent study of young adults) did not impact responses to messaging [40]. Terms such as “extra” walking or running and “avoidable” KJ/Cal were typically not noticed until participants were prompted. For a small number of participants ‘avoidable’ reminded them that they did not need to consume the product. However, for some it was noticed and interpreted as having little meaning.

“If this is avoidable, it’s like well, it’s probably better that I don’t rather than if I do”(Metro, M, L, P)

## 4. Discussion

This in-depth preliminary study of parents of school-aged children and adolescents found that participants preferred labels that they perceived to be credible, easy to understand, relevant and informative. Labels perceived most favourably across these criteria were also generally those perceived as having the most potential to change behaviour. Participants identified various ways to discount or reject some label messages, or to self-exempt from them, both from their own perspective and that of their child/adolescent. Finally, some labels were perceived as having some negative implications, such as body image concerns (e.g., obesity label) or promoting higher sugar consumption (e.g., energy information).

Across these themes and subthemes, the label that was perceived the most favourable was the text label depicting the number of teaspoons of sugar in a beverage. Minimal potential negative issues were raised by participants in relation to this label, similar to that found in a study of young adults [40], albeit the young adults in the previous study were found to exhibit a stronger negative reaction. Participants of the present study had some familiarity with this information via exposure at dental clinics and schools, which may explain their slightly tempered reaction. Regardless, the labels appeared to promote similar reactions from participants in that they were perceived as likely to prompt reconsideration of consumption for parents themselves and their children.

Use of graphics and pictograms may increase effectiveness of labels and health messages, potentially through enhancing understanding of the messages being conveyed [37,50,51,52,53]. However, many participants in this study preferred the text version of the teaspoons message for themselves, due to simplicity and ease of interpretation. The usefulness of the pictogram was noted for children as it required lower literacy, however, some indicated it created some ambiguity regarding the substance (sugar) on the teaspoon and interpretation of “16 X…”. The pictogram label is worthy of further consideration, but may require the statement ‘High in sugar’ or other refinement to reduce potential for ambiguity. Overall, parents considered the teaspoons labels to be relevant and acceptable for themselves and their children.

Label shape and use of the signal word, “warning”, facilitated perceptions of seriousness and credibility across all label types similar to previous studies [34,39,40]. Consistent with the previously mentioned study of young adults [40], comparisons were drawn with government tobacco warnings, stop sign shapes and danger warnings. Seriousness and credibility were more often noted for the rectangle shape, however the rectangle shape featured more prominently in the images used in this study and shown first to participants (see Table 1). A quantitative study comparing the same messages with different label shapes would assist in identifying the optimal shape for Australian consumers.

A preference for simple, easy to interpret information was observed in this study, consistent with previous qualitative studies of different groups of consumers [18,33,34,35,36,37,38,40]. Information that required additional context or explanation was perceived as less effective. Examples included labels that were considered too complex (KJs/Cal labels), not specific enough (“high in sugar”), lacking in context (RDI labels) or posing risk of misinterpretation. Of concern, “12% RDI” label was sometimes perceived as a positive aspect that may potentially increase consumption (noted by parents as a particular concern for children), similar to the previous study of young adults [40]. In contrast, the teaspoons label did not require context (e.g., maximum daily sugar intake), as it was inherently “known” that it portrayed an excessive amount. These results support previous findings that the interpretability of labels and messages is enhanced by using information that is specific, simple and not requiring additional context or explanation.

The finding that the “high in” label was perceived as ambiguous is noteworthy, given this label type has been implemented in Chile, with similar systems implemented in other countries [25,26,27]. Consistent with findings among Australian young adults [40], participants in this study did not perceive the “high in” warning to be useful, informative or likely to change behaviour, particularly when compared to the more specific sugar content messages (number of teaspoons). These results indicate there may be potential for increased effectiveness of such nutrient content labels by including more specific information. It should be noted that the teaspoons metric was known among these study participants, however the metric most familiar and/or useful to consumers may change with context and between countries, e.g., use of sugar cubes or sugar packets may be more applicable in some contexts.

Health effects labels have been found to be effective in previous quantitative studies [20]; however, reactions and perceptions of health effects labels in relation to SSBs have not been widely investigated using a qualitative approach. Several issues were raised by participants in this study regarding acceptability, interpretation, wording, and potential for negative reactions with respect to health effects warnings. While the health effects messages used in this study (obesity, diabetes and tooth decay) were easy to understand, they were considered well known with limited potential to attract attention. Participants preferred “contributes to” than “causes” and indicated a risk of message rejection if information was not perceived as factual. This finding is noteworthy given the perceptions of messages as factually correct and uncontroversial are likely to contribute to political feasibility [54]. Some participants perceived the health effects messages as patronising and/or authoritarian and noted the potential for “reaction” against such warnings, particularly when contrasted with the more educative teaspoons message. This was not identified in the previous study of young adults [40]; however, other studies have observed the potential for consumers to react negatively to policy options they perceive to be authoritarian and/or interfere with personal choice/freedom (e.g., a sugar tax), and react more positively towards options perceived as educative and supportive [28,29,30]. Message framing and tone are likely to contribute to the effectiveness of health effects labels, and the intended audience should be considered in terms of relevance and acceptability.

Communicating sugar consumed in beverages via exercise equivalents (amount of exercise to ‘burn off’ the sugar) has been suggested as a potential effective health message [50,55]. Exercise equivalent messages, while considered by the study participants as interesting and acceptable to children, offered scope for misinterpretation and were perceived as unlikely to change behaviour, particularly for very active individuals (children and adults). The issue of discounting health risks and information for those who appear to be low risk (healthy weight or thin, or physically active), indicates an important knowledge gap: health risks associated with overconsumption of sugar are not negated by appearing fit and healthy [56,57]. Particularly for children, overconsumption of sugar can displace essential nutrition in the diet, cause dental caries and set up lifelong unhealthy dietary habits [58,59] These findings indicate a need for clear messaging to increase understanding of health risks of overconsumption of sugar for all.

Participants in this study provided careful reflection on negative aspects of labels, particularly in relation to acceptability and suitability for their children. The obesity label raised issues of body image or promoting weight/obesity stigma. Participants raised concerns that children may choose lower sugar drinks with other perceived harmful ingredients or additives to reduce sugar consumption. Substitution effects as consequences of policies aimed at reducing SSB consumption is an area of research that requires further investigation and careful consideration. Several studies have shown a propensity for participants to switch to alternative beverages in response to policies such as taxes or labelling, both in real life and in experimental settings [33,60,61]. It is important that unintended consequences and risks are explored and considered in the implementation of labelling policies and weighed against potential positive health outcomes achieved.

## 5. Limitations

The results of this study should be interpreted within the limitations of qualitative methodology. Results are based on a sample of 82 participants and represent their subjective perceptions. There is a risk of social desirability effects, such that participants’ views can be influenced by others’ opinions in focus groups. As a qualitative study, results are not intended to be generalisable to the whole population, and perceptions of labels will likely differ across different settings and demographic subgroups. However, they do provide invaluable insight into understanding of labels of one important consumer group (parents), which can form the foundation for larger quantitative studies and further exploration among other key demographic subgroups (e.g., adolescents) and contexts. Labels were presented in a fixed order, therefore, there risk a risk of participant fatigue such that reactions may have diminished somewhat over the course of the group discussion. While overall, participants showed preference for the teaspoons label, it is unknown to what extent these perceptions would translate into behavioural change. A larger warning label evaluation study would be required to quantify these results and assist in determining which labels would lead to biggest change in behavioural outcomes.

## 6. Conclusions

Interest in warning labels on products such as sugar-sweetened beverages is growing. This study provides new insight into a group of parent consumers to help inform label development for further research and for consideration in policy development. Overall, the participants in this study had similar reactions to labels as young adults (as identified in a previous study), with teaspoons of sugar being the label perceived as having the greatest potential to change behaviour. However, their responses also highlighted there are issues to be considered dependent on the labels used and intended audience/target population if these labels were introduced (e.g., body image, misinterpretation). These reactions provide useful preliminary insights for the development and future testing of warning labels to ensure optimal message content and framing, and to inform policy development. These preliminary results point to sugar labelling as worthy of consideration for further quantitative testing, and for countries developing labels aimed at reducing consumption of sugar-sweetened beverages.

## Figures and Tables

**Table 1 nutrients-14-04173-t001:** Warning label sets, in sequential order (from 1 to 6) as they were shown to participants.

1. Health Effects	2. Sugar Content + Health Effect	3. Content Information and Energy	4. Exercise Equivalents	5. High in Sugar + Teaspoons Pictograms	6. Teaspoons Pictograms
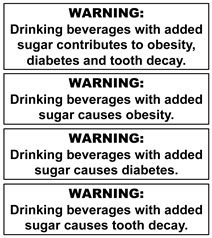	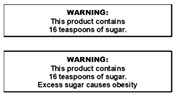	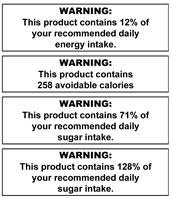	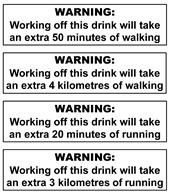	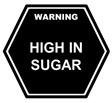 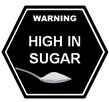 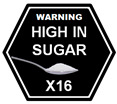	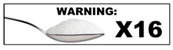 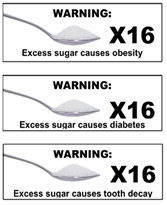

**Table 2 nutrients-14-04173-t002:** Number of participants by location, parental sex and education level, and child school level.

Child School Year Level	Groups	*n*	Sex (Parent)		Location		Parent Education Level	
			Male	Female	Metropolitan	Regional	Low	Medium -High
Primary	3	22	6	16	14	8	14	8
Primary and Secondary	5	32	17	15	14	18	16	16
Secondary	4	28	12	16	16	12	12	16
Total	12	82	35	47	44	38	42	40

**Table 3 nutrients-14-04173-t003:** Themes and subthemes.

Theme	Subthemes	Description
1. Perceptions of Label Effectiveness		Participants appraised labels on a number of criteria when considering the potential effectiveness of labels, based on the identified subthemes.
	(1a) Credibility	Participants’ perceptions of credibility.
	(1b) Informative and useful	Participants’ perceptions of a label being informative and useful.
	(1c) Personal relevance	Participants’ perceptions of personal relevance of the label (for self and child(ren)).
	(1d) Potential to change behavior	Participants’ perceptions of whether the label had potential to change behavior.
2. Opportunities for Self-Exemption and Message Rejection		Participants found opportunities for self-exemption and rejection for many labels through questioning credibility of messages (e.g., is it factual?), resistance to message ‘tone’, misinterpretation and possibility of remedial action.
3. Potential Negative Implications		Participants raised some potential negative consequences that could arise as a result of their children seeing the labels.

## Data Availability

The data are not publicly available due to ethical restrictions.

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
