# Peer review of "“No Child or Adult Would Ever Probably Choose to Have 16 Teaspoons of Sugar”: A Preliminary Study of Parents’ Responses to Sugary Drink Warning Label Options"

_nutrients, 2022, doi:10.3390/nu14194173_

Round 1

Reviewer 1 Report

The study aimed at determining parents’ perceptions and understanding of a range of warning labels for sugar-sweetened beverages.

In my opinion, the study is very interesting, the introduction is well structured and clear and highlights the gap in the literature. The text is well written and easy to read. Although the study is of a qualitative nature, it allows obtaining important food for thought for future policies.

I did not find any major or minor concerns regarding the design or the results presented and I am glad to recommend this paper for publication.

Author Response

Reviewer 1

The study aimed at determining parents’ perceptions and understanding of a range of warning labels for sugar-sweetened beverages.

In my opinion, the study is very interesting, the introduction is well structured and clear and highlights the gap in the literature. The text is well written and easy to read. Although the study is of a qualitative nature, it allows obtaining important food for thought for future policies.

I did not find any major or minor concerns regarding the design or the results presented and I am glad to recommend this paper for publication.

Response: Thank you for your positive review.

Reviewer 2 Report

I suggest to change the type of paper from "Article" to "Communication".

I sggest to insert in the title "Preliminary Study" and explain along the paper and in Conclusion that this represents a preliminary study and further researches are needed. The context in Introduction should be better structured and summarized.

Additional lines on sugar-sweetened beverages shoul be added.

The lines in Introduction from One policy option ...to progress this potential policy option should be clarified and better argumented.

The aim and the novelty character of this study should be better explained.

Major details on study design should be inserted, including a graphical scheme.

Tables should be better represented.

Explain better the limitations of studies and including in Conclusion also practical applications and further researches.

Author Response

Reviewer 2

  1. I suggest to change the type of paper from "Article" to "Communication".

Response: Thank you for reviewing our manuscript. Unfortunately, we believe this in-depth qualitative study of 12 focus groups of parents (n=82) is not suited to a short Communication. While the results will inform future work in the area, the results are also informative in their own right. As such, we have decided to keep the submission as an Article.

  1. I sggest to insert in the title "Preliminary Study" and explain along the paper and in Conclusion that this represents a preliminary study and further researches are needed. The context in Introduction should be better structured and summarized.

Response: We have not inserted “Preliminary Study” into the title, but have added “Qualitative Study” so the reader will be aware of the study design. We understand that many researchers undertake a process where a qualitative exploration is undertaken to ascertain participants’ responses prior to any quantitative testing. However, as this is a particularly large qualitative study, while results will inform future quantitative work, it is regarded as a study in its own right.

  1. Additional lines on sugar-sweetened beverages shoul be added.

Response: We have added more contextual information on the health issues and burden caused by consumption of SSBs.

  1. The lines in Introduction from One policy option ...to progress this potential policy option should be clarified and better argumented.

Response: We have moved up evidence regarding warning labels as a potential policy option into this paragraph to strengthen the argument.

  1. The aim and the novelty character of this study should be better explained.

Response: We added text to highlight the novelty of this study (middle paragraph, page 3; & in the last paragraph of the introduction). The novelty is the insight it provides beyond quantitative studies, and particularly into elements of warning labels that are perceived to be effective or not effective, it includes a large and varied range of labels (21 labels) within one study, and has been conducted among a key demographic, parents, in a new setting (Australia) to other qualitative studies.

  1. Major details on study design should be inserted, including a graphical scheme.

Response: The study design is a simple focus group study and did not involve multiple phases - we have attempted to clarify this. We have incorporated a simple graphical scheme depicting the design of this study, available as supplementary material.

  1. Tables should be better represented.

Response: It is not clear from this comment what needs to be ‘better represented’ from the tables. As such we have made some alterations that we hope may improve clarity.

  1. Explain better the limitations of studies and including in Conclusion also practical applications and further researches.

Response: We have added text and clarifications to explain the limitations of the study in ‘Limitations’, and communicate the practical application for future research (in Conclusion).

Reviewer 3 Report

The present article addresses an important public health topic. The study is well-designed and the manuscript is well-written. I have no comments and I recommend acceptance.

Author Response

Reviewer 3

The present article addresses an important public health topic. The study is well-designed and the manuscript is well-written. I have no comments and I recommend acceptance.

Response: Thank you for your positive review.

Round 2

Reviewer 2 Report

As previously indicated I suggest to change the type of paper from "Article" to "Communication". I suggest to insert in the title "Preliminary Study" and explain along the paper and in Conclusion that this represents a preliminary study and further researches are needed. 

Author Response

Reviewer 2: As previously indicated I suggest to change the type of paper from "Article" to "Communication". I suggest to insert in the title "Preliminary Study" and explain along the paper and in Conclusion that this represents a preliminary study and further researches are needed. 

Response to Reviewer: We have made the requested amendments, highlighted in yellow.